# Height & income: Labor returns of health in Mexico from 2000 to 2018

**Juan Pablo Gutiérrez**[1]*, Stefano M. Bertozzi[2,3,4]

**1** Center for Policy, Population & Health Research, School of Medicine, National Autonomous University of Mexico, Mexico City, Mexico, **2** School of Public Health, University of California, Berkeley, California, United States of America, **3** Instituto Nacional de Salud Pública, Cuernavaca, Mexico, **4** University of Washington, Seattle, Washington, United States of America

* jpgutierrez@unam.mx

## Abstract

Investment in health has been proposed as a mechanism to promote upward social mobility. Previous analyses have reported inconsistent estimates of the returns to investment in health in Mexico based on different models for different years. We aim to estimate returns for Mexico using data from four time points Adult height and labor income are drawn from the periodical national health and nutrition surveys–a group of relatively standardized surveys—that are representative of individuals living in the country in 2000, 2006, 2012 & 2018. These surveys collect anthropometric measurements and information on individuals' labor income. We estimated Mincerian models separately for men and women using OLS, Heckman, instrumental variables, and Heckman with instrumental variables models. Our results indicate significant and positive returns to health for the four surveys, similar in magnitude across years for women and with variations for men. By 2018, returns to health were about 7.4% per additional centimeter in height for females and 9.3% for males. Investments in health and nutrition during childhood and adolescence that increase health capital–measured as adult height—may promote social mobility in Mexico and similar countries to the extent that these investments differentially increase health capital among the poor.

## Background

From a human capital perspective, investments in health contribute to social mobility by increasing the healthy time available for economic activities and studying and enhancing cognitive abilities development [1]. From this perspective, health—in addition to being itself a desirable condition—is an investment that enhances school performance and increases individual physical capacities, enhancing skills that will increase the likelihood of higher-income.

While there are standard indicators of health status, these measures generally refer to the current health (at the time of measurement). That is, establishing, for example, the personal assessment of the health status ("how do you rate your health?"), frames the question at a particular moment that is affected by dynamic conditions, which include the presence of seasonal infections, accidents, and similar events.

**Data Availability Statement:** The datasets analyzed during the current study are available in the ENSANUT repository at http://ensanut.insp.mx.

**Competing interests:** The authors have declared that no competing interests exist.

In order to capture the accumulated health (capital health), a metric that captures the level of health achieved in a certain period seems desirable. Based on the growth and development of individuals, the literature has proposed the use of the maximum height reached as a measure of accumulated health. Although it reflects genetic variations, the rationale is that height is also the result of exposure to diseases during growth; the nutrition received, hygienic and feeding practices, and behaviors that affect health [2,3].

The literature has already discussed that greater variability in height independent of genetic aspects is observed in medium and low-income contexts, while in high-income settings, most of the variation in height is related to genetic aspects [4–6].

Adults' height as a measure of health capital was initially reported in historical studies that look at the evolution in height as a measure of the development of the countries, as Fogel proposed [7].

In addition, several studies have already analyzed the relationship between height and income as an approach to estimating the labor returns on investment in health, identifying that they are positive and significant in low and middle-income countries [5].

For Mexico, while there are previous estimations on the health returns using height, results are not consistent: while an analysis using data from the National Health Survey 2000 found rather modest returns [8], in a previous analysis from our group, we reported higher returns using the National Health & Nutrition Survey 2006 [9], both estimations using cross-sectional data.

Using longitudinal data from a program that covered households in the bottom of the income distribution, we estimated health returns more consistent with our previous 2006 [10].

One potential explanation for this difference is that we decided to instrument height recognizing the potential endogeneity that has been discussed previously [1], that is, potential unobserved differences in parental preferences and individual endowment that could affect labor income. An alternative explanation is that health returns may change over time, so differences using data from different years are expected.

In order to address those two issues, in this analysis, we estimated the labor income returns of health in Mexico for 2000, 2006, 2012, and 2018 using our previous approach to instrument height and measuring how those returns varied over the analytical period.

## Methods

This study uses a repeated cross-sectional analysis with data from the National Health Survey 2000, the National Health and Nutrition Survey 2006, the National Health and Nutrition Survey 2012, and the National Health and Nutrition Survey 2018, four probabilistic and representative surveys of Mexico's population living in private dwellings. Details on surveys methodology have been published elsewhere; the four surveys used similar procedures for sampling, that is, the national geostatistical framework from the national statistics institute, and questionnaires for the later rounds were based on the previous ones, as described in their methodological reports [11–14].

### Data

The total sample size collected for the ENSA/ENSANUT surveys is 45,870, 46,000, 50,528, and 44 064 households, respectively for 2000, 2006, 2012 & 2018. Data were collected from each household using a set of instruments: a household questionnaire that collect demographic and socioeconomic characteristics as well as dwelling characteristics for all household members; individual questionaries for selected individual in each age-group (children, adolescents, and adults); a questionnaire on health services utilization for selected individuals; and

measurement forms to capture anthropometric data collected from selected individuals by qualified personnel.

For this analysis, we use data from the four surveys from both self-reported variables (from the individual adult and household questionnaires) and the anthropometric data (individuals' weight in kg and height in cm) measured by trained personnel.

Each survey round gathered data on individuals´ age, sex, education (schooling years completed at the time of the interview), marital status (categorized as united or not united at the time of the interview), for women if they had had children and if they have a son or daughter aged five years or less at the time of the survey, employment status (considering as working at the time of the interview to those who reported having a job—whether or not they had worked that week—, or that they carried out some other economic activity) and earnings. Data on earnings (that includes both wages and self-employment income) are self-reported in the individual survey. They are collected bythe period in which the income was received (weekly, biweekly, monthly, annually) and then standardized as weekly income in pesos. For comparison, average income for 2000, 2006 and 2012 are also reported in 2018 pesos by adjusting for inflation using the national consumer price index [15].

In addition, we generated a socioeconomic status index of the household in deciles for each survey, using an imputation from the Household Income and Expenditure Survey closest to the health survey year. The detail of the imputation is reported elsewhere but, in summary, is implemented using variables of household structure, characteristics of the dwelling, and household assets (available both in the respective ENIGH and in the respective health survey), to predict the household income level from the national distribution [16,17]. Using the estimated household income, we defined deciles of this variable for those adults in each survey.

Using the anthropometric measurements for weight and height, we calculated each individual's body mass index (BMI) as kilograms per meter squared of height.

To analyze individuals who have already reached their maximum height and whose height decline has not started yet, we implemented the analysis for those 25 to 45 years of age [18,19].

## Analysis

To estimate the labor income returns to health, we used a Mincerian model [20], with an additional correction for the selection bias in terms of labor participation on which labor income depends, using the approach proposed by Heckman [21]. The model estimates the changes in labor income associated with changes in the health indicator, in this case, height. The general model has the form expressed in Eq (1):

$$\ln W_i = f(S_i, E_i, Ex_i, X_i) \tag{1}$$

where $\ln W_i$ is the natural logarithm of individual *I's* monthly income. *W* is a function of the levels of health ($S_i$), education ($E_i$), experience ($Ex_i$), and other individual characteristics ($X_i$) [8,10,22–24]. We estimated experience as the potential years of work as age minus the schooling years and minus 6 years (under the assumption that school starts at 6 years), as suggested in previous studies [25], but it was not included in the estimations as multicollinearity with age was identified. We were not able to include an indication of ability as this was not included in the surveys.

The Heckman approach seeks to correct for the censorship observed in labor income (given by the non-random labor participation) by including the probability of obtaining labor (or self-employment) income in the estimate. To estimate the selection equation, in addition to sex, age, BMI, and schooling, we added the identification of whether the person has a partner and the socioeconomic decile. Having a partner may affect the decision to participate in

the labor market by having an explicit intra-household division of labor, that is, one partner participating in the labor market and the other not. The same applies to having children, which may affect the decision to participate or not in the labor market, particularly for women who may decide to stay at home. The inclusion of the household socioeconomic decile is related to the differential need to participate in the labor market, as those at the bottom depending on labor earnings, but those at the top may have other sources of income.

To address the potential endogeneity between height and income, we instrumented height using the average height among individuals living in the same census tract, excluding the individual own value. This instrument rely on the literature that shows spatial correlation on height; an individuals' height is correlated with the height of others living in the same neighborhood [26,27]. Use of this instrument is predicated on the assumption that the primary mechanism through which average community height is associated with an individual's income is via the association with that individual's height [28].

Given differential labor force participation by sex, separate models were estimated for men and women. To estimate the selection equation, we added as variables both the identification of whether the person has a partner and for women if they have children and if so, if they have a child under five years (only for women due to data availability).

Thus, the estimated model is:

$$\log(W_i) = \beta \hat{S}_i + \varphi X_i + \varepsilon_i \tag{2}$$

$$\dot{E}_i = \gamma Z_i + u_i \tag{3}$$

where $\log(W_i)$ is the natural logarithm of the monthly income reported by the individual $i$, $\hat{S}_i$ is the value of the health indicator, in this case the natural logarithm of the height—so the coefficient $\beta$ is an elasticity, $X_i$ a vector of individual characteristics (sex, age, age squared, schooling years category), $\dot{E}_i$ is the identification of whether the person generates earned income (labor participation), and $Z_i$ the individual characteristics that determine labor participation (being in a relationship, BMI and SES decile).

Heckman models for each survey were implemented using Stata 16.1 command Heckman with full maximum likelihood option. The IV models were implemented using the ivregress command in Stata 16.1

We estimated the models for each round separately. All results reported here are weighted using the specific weights for each survey as well as the survey design.

### Ethics approval and consent to participate

Public available de-identified data were used for this analysis; that is, this is a secondary analysis of national surveys collected by the National Institute of Public Health. All four surveys reported using informed consent and ethical procedures approved by the INSP´s IRB [11–14].

### Results

The descriptive statistics of the population (adults aged 25 to 45 years with anthropometric data) are reported in Table 1 for the four surveys that is, ENSA 2000, ENSANUT 2006, ENSANUT 2012, and ENSANUT 2018. While for the ENSA 2000 of all adults with individual questionnaires, 46% were males, for the latter surveys, this percentage decreased to 39% in 2006, 41% in 2012, and 37% in 2018. It is not clear whether men were more likely to be home in 2000 or if the field teams made greater efforts to return to administer a questionnaire to men. Women are more likely to be home when the household is visited by the survey team, as

**Table 1. Means or percentages (95% CI) of individuals aged 25 to 45, by survey year and sex.**

| | 2000 | | |
|---|---|---|---|
| **VARIABLES** | **All** | **Females** | **Males** |
| Sex (% males) | 46% | | |
| | (44% - 47%) | | |
| Age | 33.75 | 33.73 | 33.78 |
| | (33.63–33.88) | (33.59–33.86) | (33.57–33.99) |
| Schooling years | 7.96 | 7.39 | 8.63 |
| | (7.78–8.14) | (7.21–7.58) | (8.40–8.86) |
| Non-schooling (%) | 6% | 6% | 6% |
| | (4% - 7%) | (4%- 7%) | (4%- 7%) |
| Speaks an indigenous language (%) | 6% | 6% | 7% |
| | (5% - 8%) | (5% - 8%) | (5% - 9%) |
| Height (cm) | 159.58 | 153.98 | 166.22 |
| | (159.26–159.90) | (153.75–154.21) | (165.79–166.65) |
| Weight (kg) | 69.65 | 65.91 | 74.08 |
| | (69.23–70.07) | (65.50–66.32) | (73.45–74.71) |
| Body mass index (BMI) | 27.31 | 27.78 | 26.75 |
| | (27.19–27.43) | (27.63–27.93) | (26.55–26.95) |
| BMI < 25 (Normal) % | 35% | 33% | 37% |
| | (34% - 36%) | (32% - 34%) | (35% - 39%) |
| BMI > = 25 & < 30 (Overweight) % | 41% | 38% | 44% |
| | (40% - 42%) | (37% - 39%) | (42% - 46%) |
| BMI > 30 (Obese) % | 24% | 29% | 19% |
| | (23% - 25%) | (28% - 30%) | (17% - 20%) |
| Work (%) | 57% | 29% | 91% |
| | (56% - 59%) | (28% - 31%) | (89% - 92%) |
| Monthly income | 3,245 | 2,539 | 3,517 |
| | (2,952–3,539) | (2,390–2,689) | (3,127–3,908) |
| Montly income in 2018 pesos | 7,223 | 5,652 | 7,829 |
| Have a child (%) | | 89% | |
| | | (88% - 90%) | |
| Have a child under 5 years old (%) | | 44% | |
| | | (42% - 45%) | |
| Living with a partner (%) | 81% | 82% | 79% |
| | (79% - 82%) | (80% - 83%) | (78% - 81%) |
| | 2006 | | |
| **VARIABLES** | **All** | **Females** | **Males** |
| Sex (% males) | 39% | | |
| | (37% - 40%) | | |
| Age | 34.84 | 34.83 | 34.86 |
| | (34.68–35.00) | (34.62–35.04) | (34.61–35.10) |
| Schooling years | 6.67 | 6.48 | 6.97 |
| | (6.54–6.81) | (6.32–6.64) | (6.77–7.18) |
| Non-schooling (%) | 16% | 16% | 17% |
| | (15% - 17%) | (14% - 17%) | (15% - 19%) |
| Speaks an indigenous language (%) | 7% | 8% | 6% |
| | (6% - 8%) | (6% - 9%) | (5% - 7%) |
| Height (cm) | 158.45 | 153.42 | 166.45 |

(*Continued*)

**Table 1.** (Continued)

| | | | |
|---|---|---|---|
| | (158.20–158.70) | (153.21–153.64) | (166.13–166.76) |
| Weight (kg) | 70.89 | 67.59 | 76.14 |
| | (70.48–71.29) | (67.14–68.03) | (75.52–76.75) |
| Body mass index (BMI) | 28.20 | 28.68 | 27.44 |
| | (28.06–28.34) | (28.51–28.86) | (27.24–27.63) |
| BMI < 25 (Normal) % | 29% | 27% | 31% |
| | (27% - 30%) | (25% - 29%) | (29% - 33%) |
| BMI > = 25 & < 30 (Overweight) % | 41% | 38% | 45% |
| | (39% - 42%) | (37% - 40%) | (43% - 47%) |
| BMI > 30 (Obese) % | 31% | 35% | 24% |
| | (29% - 32%) | (33% - 36%) | (22% - 26%) |
| Work (%) | 61% | 40% | 95% |
| | (60% - 63%) | (39% - 42%) | (94% - 96%) |
| Monthly income | 3,878 | 3,209 | 4,332 |
| | (3,753–4,003) | (3,058–3,360) | (4,166–4,497) |
| Montly income in 2018 pesos | 6,588 | 5,452 | 7,359 |
| Have a child (%) | | 82% | |
| | | (80% - 83%) | |
| Have a child under 5 years old (%) | | 40% | |
| | | (38% - 41%) | |
| Living with a partner (%) | 79% | 77% | 81% |
| | (77% - 80%) | (76% - 79%) | (79% - 82%) |
| | | **2012** | |
| **VARIABLES** | **All** | **Females** | **Males** |
| Sex (% males) | 41% | | |
| | (40% - 42%) | | |
| Age | 34.56 | 34.49 | 34.67 |
| | (34.40–34.73) | (34.29–34.69) | (34.41–34.94) |
| Schooling years | 8.61 | 8.41 | 8.90 |
| | (8.48–8.75) | (8.25–8.57) | (8.71–9.10) |
| Non-schooling (%) | 6% | 7% | 5% |
| | (6% - 7%) | (6% - 8%) | (6% - 7%) |
| Speaks an indigenous language (%) | 7% | 7% | 6% |
| | (6% - 7%) | (6% - 8%) | (5% - 7%) |
| Height (cm) | 159.16 | 153.70 | 167.06 |
| | (158.89–159.42) | (153.47–153.93) | (166.77–167.35) |
| Weight (kg) | 72.40 | 68.23 | 78.45 |
| | (72.00–72.81) | (67.77–68.69) | (77.84–79.06) |
| Body mass index (BMI) | 28.53 | 28.86 | 28.05 |
| | (28.39–28.66) | (28.69–29.03) | (27.86–28.24) |
| BMI < 25 (Normal) % | 27% | 27% | 27% |
| | (25% - 28%) | (25% - 28%) | (25% - 29%) |
| BMI > = 25 & < 30 (Overweight) % | 41% | 38% | 44% |
| | (39% - 42%) | (36% - 40%) | (42% - 47%) |
| BMI > 30 (Obese) % | 33% | 36% | 29% |
| | (31% - 34%) | (34% - 37%) | (26% - 31%) |
| Work (%) | 62% | 42% | 90% |
| | (60% - 63%) | (40% - 44%) | (88% - 91%) |

(*Continued*)

**Table 1.** (Continued)

| | All | Females | Males |
|---|---|---|---|
| Monthly income | 4,923 | 4,662 | 5,100 |
| | (4,366–5,480) | (3,316–6,008) | (4,912–5,288) |
| Montly income in 2018 pesos | 6,468 | 6,125 | 6,700 |
| Have a child (%) | | 83% | |
| | | (82% - 84%) | |
| Have a child under 5 years old (%) | | 35% | |
| | | (33% - 36%) | |
| Living with a partner (%) | 77% | 76% | 78% |
| | (75% - 78%) | (74% - 77%) | (76% - 80%) |
| | | **2018** | |
| **VARIABLES** | **All** | **Females** | **Males** |
| Sex (% males) | 37% | | |
| | (34% - 40%) | | |
| Age | 34.91 | 34.77 | 35.14 |
| | (34.53–35.29) | (34.26–35.28) | (34.63–35.65) |
| Schooling years | 9.32 | 9.37 | 9.23 |
| | (8.98–9.66) | (8.93–9.80) | (8.71–9.76) |
| Non-schooling (%) | 2% | 2% | 1% |
| | (1% - 2%) | (1% - 2%) | (1% - 2%) |
| Speaks an indigenous language (%) | 10% | 10% | 11% |
| | (7% - 14%) | (6% - 13%) | (4% - 19%) |
| Height (cm) | 158.76 | 154.09 | 166.70 |
| | (158.01–159.50) | (153.37–154.80) | (165.53–167.86) |
| Weight (kg) | 72.96 | 69.40 | 79.03 |
| | (71.76–74.17) | (67.86–70.95) | (77.58–80.47) |
| Body mass index (BMI) | 28.89 | 29.18 | 28.40 |
| | (28.54–29.25) | (28.70–29.66) | (27.89–28.92) |
| BMI < 25 (Normal) % | 21% | 21% | 22% |
| | (18% - 25%) | (18% - 25%) | (16% - 27%) |
| BMI > = 25 & < 30 (Overweight) % | 40% | 38% | 43% |
| | (35% - 45%) | (33% - 42%) | (33% - 52%) |
| BMI > 30 (Obese) % | 39% | 41% | 36% |
| | (33% - 45%) | (35% - 46%) | (24% - 48%) |
| Work (%) | 61% | 44% | 91% |
| | (58% - 65%) | (39% - 48%) | (89% - 93%) |
| Monthly income | 6,852 | 6,110 | 7,456 |
| | (6,384–7,321) | (5,454–6,766) | (6,758–8,154) |
| Montly income in 2018 pesos | 6,852 | 6,110 | 7,456 |
| Have a child (%) | | 84% | |
| | | (81% - 87%) | |
| Have a child under 5 years old (%) | | 30% | |
| | | (26% - 33%) | |
| Living with a partner (%) | 76% | 75% | 77% |
| | (74% - 79%) | (72% - 79%) | (74% - 81%) |

Source: Own estimates based on ENSA 2000, ENSANUT 20006, ENSANUT 2012, and ENSANUT 2018.

suggested by their differential labor force participation. The average age among the population analyzed (25–45 years of age) is expected to remain similar over time and was 33.75 years in 2000, 34.84 years in 2006, 34.56 years in 2012, and 34.91 years in 2018.

One relevant difference over time is reported for years of schooling: while average years of schooling increased—as expected—during the period going from 8.04 years in 2000 to 9.32 years in 2018, there was a decrease between 2000 and 2006. In 2006 the ENSANUT over-sampled individuals from poor areas, which was only partially addressed by the survey weights.

Height remained relatively constant over the years, with females averaging 153.98 cm in 2000, 153.42 cm in 2006, 153.70 cm in 2012, and 154.09 cm in 2018. For males, the average height was 166.22 cm, 166.45 cm, 167.06 cm, and 166.70 cm for the same years, respectively.

In contrast, there is a significant increase in average weight, leading to a rise in the percentage of overweight and obese individuals (body mass index greater than 25.0), with reached 78% in 2018, up from 65% in 2000. The average body mass index (BMI) for females increased from 27.78 in 2000 to 28.68 in 2006, 28.86 in 2012, and 29.18 in 2018. For males, the increase was from 26.75 in 2000 to 27.44 in 2006, 28.05 in 2012, and 28.40 in 2018.

The percentage of individuals reporting having a job or being self-employed remained nearly constant: 57% in 2000, 61% in 2006, 62% in 2012, and 61% in 2018, with a notable increase in female participation: from 29% in 2000 to 44% in 2018. Although there was a 5.0% decrease in monthly income (in 2018 pesos) from 2000 to 2018, the decline was observed only among males (-4.8%), while females experienced an 8.1% increase. Despite this, the average income gap remained positive for males, with men earning 22% more than women in 2018, compared to a 38% difference in 2000.

Fig 1 illustrates the average height among Mexicans 25 to 45 years of age by socioeconomic deciles and sex, for the years 2000, 2006, 2012, and 2018. A fact stands out from this figure: consistently, over time, there is a positive relationship between height and socioeconomic decile. Additionally, the data depict in the figure also suggest a widening height gap between income deciles throughout the analyzed period.

Specifically, for females, the height gap between the first and tenth deciles was 4.13 cm in 2000, increasing to 5.68 cm by 2018, indicating a growth of 1.55 cm (p = 0.0973). Conversely, for males, the gap expanded from 6.03 cm in 2000 to 9.65 cm in 2018, representing a notable increase of 3.61 cm (p = 0.0253).

Considering that the estimation of returns to health refers to the change in income for a 1% change in height, it is important to keep in mind that the differences in height between the lowest and highest deciles for females went from 2.77% in 2000 to 3.46% in 2006, 4.81% in 2012, and 5.16% in 2018, while for males they went from 4.25% in 2000, 4.28% in 2006, 4.74% in 2012, and 5.09% in 2018. That is, the gap for females almost doubled in the period and increased by about 20% for males in the same period.

## Instruments

The log height was tested for endogeneity, rejecting the null hypothesis of exogeneity. To instrument this variable, we used the average height from those individuals living in the same census tract. Previous literature has reported a spatial correlation in height; that is, the height of individuals from the same neighborhood is correlated [26,27]. The mechanism of this correlation could be related to social clustering, that is, individuals from similar social backgrounds living in the same neighborhoods.

This suggests that the average of the neighborhood (excluding the individual´s own height) is an adequate instrument.

# Mean Height with Confidence Intervals by Year and Sex

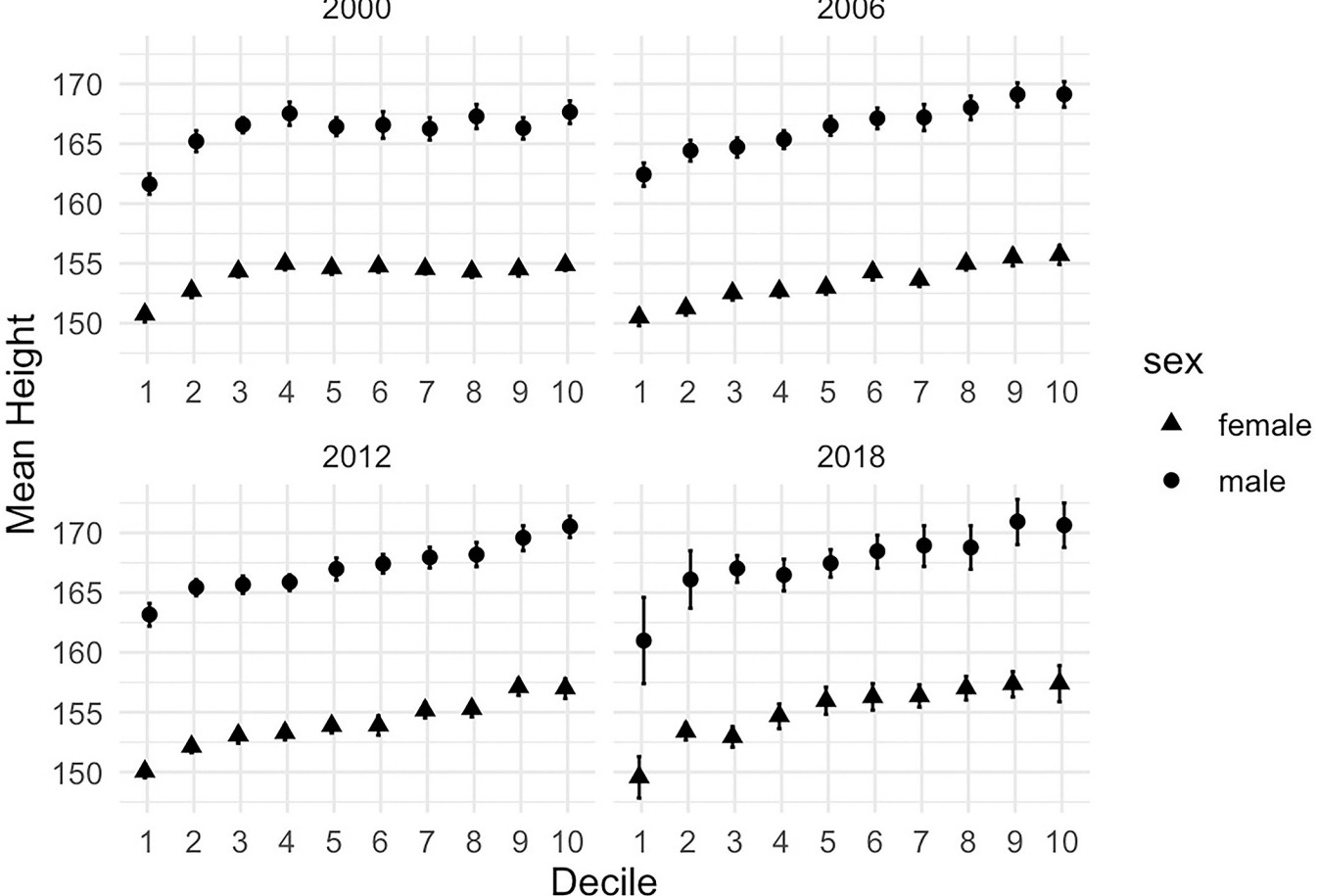

**Fig 1. Average height in centimeters of people aged 25 to 45 years in Mexico for the years 2000, 2006, and 2012, by sex and income decile.** Source: ENSA 2000, ENSANUT 2006, 2012, & 2018.

The results for the first stage of the instrumentation are reported in Table 2. The estimated F of the model suggests that we have a strong instrument.

## Labor income returns of health

We report the full set of estimations in Table 3 (OLS, IV, Heckman, and IV+Heckman) for 2000, 2006, 2012 & 2018. Overall, OLS estimations are lower than IV and IV+Heckman, and close to the Heckman estimations. Additionally, estimated returns tend to be higher for males than females.

For females, OLS estimations ranged from a 2.17% increase in labor income for each 1% increase in height in 2000 to a 1.42% increase in labor income in 2018 for the same increase in height. The IV estimation of returns ranged from a 10.17% increase in labor income by each 1% increase in height in 2000 to 5.8% in 2018. The Heckman estimation resulted in a 2.11% increase in labor income by each 1% increase in height in 2000 and a 1.26% increase in labor income by the same 1% increase in height by 2018. Finally, the IV+Heckman estimation reported returns of 9.64% in 2000 and 3.16% in 2018.

**Table 2. Coefficients and standard errors of the first stage of IV model by year.**

| Variables | 2000 | 2006 | 2012 | 2018 |
|---|---|---|---|---|
| Average height | 0.77*** | 0.56*** | 0.46*** | 0.39*** |
| | -0.03 | -0.02 | -0.02 | -0.04 |
| Age | 0.00 | 0.00 | 0.00** | 0.00* |
| | 0.00 | 0.00 | 0.00 | 0.00 |
| Age squared | 0.00 | 0.00 | 0.00 | 0.00 |
| | 0.00 | 0.00 | 0.00 | 0.00 |
| Sex | 0.02*** | 0.04*** | 0.04*** | 0.05*** |
| | 0.00 | 0.00 | 0.00 | 0.00 |
| Indigenous | -0.02*** | -0.02*** | -0.03*** | -0.03*** |
| | 0.00 | 0.00 | 0.00 | 0.00 |
| Urban | 0.00* | 0.00** | 0.00 | -0.01*** |
| | 0.00 | 0.00 | 0.00 | 0.00 |
| Metropolitan | | 0.00*** | 0.01*** | |
| | | 0.00 | 0.00 | |
| Constant | 1.21*** | 2.23*** | 2.73*** | 3.14*** |
| | -0.13 | -0.11 | -0.12 | -0.22 |
| Observations | | | | |
| R-squared | 0.50 | 0.53 | 0.53 | 0.54 |
| Wu-Hausman test for endogeneity | 55.22 | 51.53 | 10.06 | 51.17 |
| p | <0.001 | <0.001 | 0.00 | <0.001 |
| Partial R-squared | 0.10 | 0.09 | 0.07 | 0.07 |
| F | 408.73 | 561.26 | 366.63 | 122.25 |

Regarding education returns, in 2000, all models produced coefficients between 0.05 & 0.07, that is the percentual increase in income by each additional year of schooling was between 5% & 7%. In 2006, the estimated returns for education were around 2% to 2.6% increase for each additional schooling year using. By 2012, returns for schooling years ranged from negative (-2.2%) to positive 1.85%, with the IV model being no significant. For 2018, returns were between 5.4% & 6.9% larger income for each additional schooling year.

For males, the OLS model reported returns in 2000 of 2.79% higher labor income by each 1% increase in height, 2.61 in 2006, non-significant in 2012, and 2.42% in 2018. In turn, the IV model estimated an increase of 11.48% in labor income by each 1% increase in height in 2000, 8.98% in 2006, 13.81% in 2012, and 8.79% in 2018. The Heckman model estimated returns of 2.79% higher labor income by each 1% increase in height in 2000, 2.56% in 2006, non-significant in 2012, and 2.45% increase in labor income for a 1% increase in height in 2018. Finally, the IV+Heckman approach produced an estimated return of 11.05% higher labor income by each 1% increase in height in 2000, then 8.39% in 2006, 11.22% in 2012, and 6.46% in 2018.

As for the education returns, for males, in 2000 the estimation ranged from 4.1% to 6.2% increase in income for each additional schooling year, while for 2006 ranged between 1.6% and 2.4%. By 2012, schooling years were no significant in any of the estimations, and then by 2018 estimations on school year returns ranged from 3.7% to 5.4%.

As discussed before, height has remained almost constant during the period (between 158cm and 159cm), so a 1% increase in height for all years represents about 1.6 cm. During the same period, the minimum wage, in 2018 pesos, has also remained quite stable, being $2,565 pesos in 2000 and $2,687 pesos in 2018, an increase of only 4.7% [29].

**Table 3. Returns on monthly labor income for height and years of study, by survey year and sex.**

| | Females | | | | | | | |
|---|---|---|---|---|---|---|---|---|
| | 2000 | | | | 2006 | | | |
| | OLS | IV | Heckman | IV+Heckman | OLS | IV | Heckman | IV+Heckman |
| log Height | 2.17*** | 10.18*** | 2.12*** | 9.64*** | 3.34*** | 9.53*** | 3.23*** | 9.27*** |
| | 0.38 | 1.60 | 0.41 | 1.26 | 0.50 | 1.87 | 0.51 | 1.83 |
| Schooling years | 0.07*** | 0.05*** | 0.07*** | 0.07*** | 0.03*** | 0.02*** | 0.02*** | 0.03*** |
| | 0.00 | 0.01 | 0.00 | 0.00 | 0.00 | 0.00 | 0.00 | 0.00 |
| BMI | -0.01** | 0.00 | -0.01** | -0.01*** | 0.00 | 0.00 | 0.00 | 0.00 |
| | 0.00 | 0.00 | 0.00 | 0.00 | 0.00 | 0.00 | 0.00 | 0.00 |
| Age | 0.08** | 0.07* | 0.09** | 0.11*** | -0.02 | -0.02 | -0.02 | -0.04 |
| | 0.04 | 0.04 | 0.04 | 0.04 | 0.05 | 0.05 | 0.05 | 0.05 |
| Age squared | 0.00 | 0.00 | 0.00** | 0.00** | 0.00 | 0.00 | 0.00 | 0.00 |
| | 0.00 | 0.00 | 0.00 | 0.00 | 0.00 | 0.00 | 0.00 | 0.00 |
| Indigenous language speaker | -0.16** | 0.14 | -0.16** | 0.08 | -0.19** | 0.06 | -0.20** | -0.01 |
| | 0.07 | 0.10 | 0.07 | 0.08 | 0.08 | 0.11 | 0.08 | 0.10 |
| Rho | | | -0.12** | -0.13** | | | -0.11 | 0.00 |
| | | | 0.06 | 0.06 | | | 0.14 | 0.11 |
| Sigma | | | -0.27*** | -0.28*** | | | -0.17*** | -0.16*** |
| | | | 0.02 | 0.02 | | | 0.03 | 0.03 |
| R2 | 0.2535 | 0.1112 | | | 0.1189 | 0.0396 | | |
| Observations | 4,760 | 4,760 | 14,527 | 14,527 | 3,874 | 3,874 | 9,229 | 9,229 |
| | Females | | | | | | | |
| | 2012 | | | | 2018 | | | |
| | OLS | IV | Heckman | IV+Heckman | OLS | IV | Heckman | IV+Heckman |
| log Height | 1.85*** | 11.51*** | 1.58** | 1.47 | 1.47*** | 7.19** | 1.38** | 3.16* |
| | 0.71 | 4.05 | 0.73 | 3.76 | 0.54 | 3.35 | 0.54 | 1.83 |
| Schooling years | 0.02** | 0.00 | -0.02** | -0.02* | 0.07*** | 0.05*** | 0.05*** | 0.06*** |
| | 0.01 | 0.01 | 0.01 | 0.01 | 0.01 | 0.01 | 0.01 | 0.01 |
| BMI | 0.00 | 0.01 | -0.01 | -0.01 | 0.00 | 0.00 | 0.00 | 0.00 |
| | 0.00 | 0.00 | 0.01 | 0.01 | 0.00 | 0.00 | 0.00 | 0.00 |
| Age | -0.01 | -0.01 | 0.02 | 0.02 | 0.06 | 0.05 | 0.08 | 0.03 |
| | 0.02 | 0.02 | 0.02 | 0.02 | 0.05 | 0.06 | 0.05 | 0.05 |
| Age squared | 0.00 | 0.00 | 0.00 | 0.00 | 0.00 | 0.00 | 0.00 | 0.00 |
| | 0.00 | 0.00 | 0.00 | 0.00 | 0.00 | 0.00 | 0.00 | 0.00 |
| Indigenous language speaker | -0.03 | 0.33* | -0.02 | 0.27* | 0.10 | 0.28 | 0.12 | 0.18 |
| | 0.09 | 0.18 | 0.07 | 0.15 | 0.12 | 0.18 | 0.11 | 0.14 |
| Rho | | | -1.87*** | -1.91*** | | | -0.49*** | -0.45*** |
| | | | 0.15 | 0.14 | | | 0.14 | 0.12 |
| Sigma | | | 0.52*** | 0.53*** | | | -0.33*** | -0.36*** |
| | | | 0.04 | 0.04 | | | 0.04 | 0.04 |
| R2 | 0.0140 | | | | 0.1652 | | | |
| Observations | 3,910 | 3,909 | 9,793 | 9,783 | 1,778 | 1,477 | 3,729 | 3,758 |
| | Males | | | | | | | |
| | 2000 | | | | 2006 | | | |
| | OLS | IV | Heckman | IV+Heckman | OLS | IV | Heckman | IV+Heckman |
| log Height | 2.79*** | 11.48*** | 2.79*** | 11.05*** | 2.61*** | 8.98*** | 2.56*** | 8.39*** |
| | 0.33 | 1.47 | 0.34 | 1.07 | 0.29 | 1.14 | 0.30 | 0.90 |
| Schooling years | 0.06*** | 0.04*** | 0.06*** | 0.06*** | 0.02*** | 0.02*** | 0.02*** | 0.02*** |

*(Continued)*

**Table 3.** (Continued)

| | | | | | | | | |
|---|---|---|---|---|---|---|---|---|
| | 0.00 | 0.01 | 0.01 | 0.00 | 0.00 | 0.00 | 0.00 | 0.00 |
| BMI | 0.01*** | 0.02*** | -0.01** | 0.01*** | 0.01*** | 0.01*** | 0.01*** | 0.01*** |
| | 0.00 | 0.00 | 0.00 | 0.00 | 0.00 | 0.00 | 0.00 | 0.00 |
| Age | 0.04 | 0.05 | 0.09** | 0.05 | 0.01 | -0.01 | -0.01 | 0.00 |
| | 0.03 | 0.03 | 0.04 | 0.03 | 0.02 | 0.03 | 0.02 | 0.02 |
| Age squared | 0.00 | 0.00 | 0.00** | 0.00 | 0.00 | 0.00 | 0.00 | 0.00 |
| | 0.00 | 0.00 | 0.00 | 0.00 | 0.00 | 0.00 | 0.00 | 0.00 |
| Indigenous language speaker | -0.33*** | -0.04 | -0.16** | -0.03 | -0.28*** | -0.05 | -0.29*** | -0.08 |
| | 0.07 | 0.08 | 0.07 | 0.06 | 0.05 | 0.07 | 0.04 | 0.05 |
| Rho | | | -0.52*** | -0.50*** | | | -0.74*** | -0.72*** |
| | | | 0.08 | 0.08 | | | 0.13 | 0.15 |
| Sigma | | | -0.37*** | -0.37*** | | | -0.51*** | -0.51*** |
| | | | 0.03 | 0.02 | | | 0.02 | 0.02 |
| R2 | 0.3323 | 0.1196 | | | 0.2304 | 0.0581 | | |
| Observations | 5,280 | 5,280 | 5,668 | 5,668 | 5,397 | 5,396 | 5,524 | 5,523 |
| | Males | | | | | | | |
| | 2012 | | | | 2018 | | | |
| | OLS | IV | Heckman | IV+Heckman | OLS | IV | Heckman | IV+Heckman |
| log Height | 1.26 | 13.82*** | 0.88 | 11.22*** | 2.11*** | 8.50*** | 2.31*** | 6.46*** |
| | 1.03 | 3.88 | 0.96 | 2.69 | 0.40 | 2.29 | 0.41 | 1.13 |
| Schooling years | 0.00 | -0.02 | 0.00 | 0.00 | 0.05*** | 0.03*** | 0.05*** | 0.05*** |
| | 0.01 | 0.01 | 0.01 | 0.01 | 0.01 | 0.01 | 0.01 | 0.01 |
| BMI | 0.00 | 0.00 | 0.00 | 0.00 | 0.01** | 0.01** | 0.01** | 0.01** |
| | 0.01 | 0.01 | 0.01 | 0.01 | 0.00 | 0.01 | 0.00 | 0.00 |
| Age | 0.08 | 0.07 | 0.02 | 0.03 | 0.03 | 0.00 | 0.01 | -0.02 |
| | 0.11 | 0.12 | 0.08 | 0.08 | 0.04 | 0.05 | 0.04 | 0.05 |
| Age squared | 0.00 | 0.00 | 0.00 | 0.00 | 0.00 | 0.00 | 0.00 | 0.00 |
| | 0.00 | 0.00 | 0.00 | 0.00 | 0.00 | 0.00 | 0.00 | 0.00 |
| Indigenous language speaker | -0.14* | 0.31* | -0.17** | 0.18 | -0.13* | 0.15 | -0.13** | 0.24*** |
| | 0.08 | 0.16 | 0.07 | 0.12 | 0.07 | 0.11 | 0.06 | 0.07 |
| Rho | | | -2.18*** | -2.21*** | | | -0.75*** | -0.78*** |
| | | | 0.08 | 0.09 | | | 0.12 | 0.12 |
| Sigma | | | 0.49*** | 0.48*** | | | -0.47*** | -0.53*** |
| | | | 0.05 | 0.05 | | | 0.05 | 0.04 |
| R2 | 0.0057 | | | | 0.1887 | | | |
| Observations | 4,854 | 4,848 | 5,335 | 5,319 | 2,138 | 2,001 | 2,330 | 2,344 |

Heckman: Selection model includes sex, being in a relationship, having children, having a child under 5 years of age, BMI and SES decile.

OLS: Controlling for age, sex & BMI.

Source: Own estimates based on ENSA 2000, ENSANUT 20006, ENSANUT 2012, and ENSANUT 2018.

The reported health returns implied that a 1% increase in height for females using the OLS estimates represented 4.79% of the minimum wage in 2000 and 3.25% in 2018, and 21.25% and 7.20% using the Heckman + IV estimates, respectively. For males, using the OLS model estimated increases, it represents 8.52% in 2000 and 7.05% in 2018 of the minimum wage, while with the IV+Heckman represents 32.18% of the minimum wage in 2000 and 17.93% in 2018.

## Discussion

The analysis demonstrates a positive relationship between the maximum height of individuals, an indicator of accumulated health during growth and development, and the labor income of adults in Mexico from 2000 to 2018. Although the magnitude varied over time and was different by sex, the relationship remained in general consistently significant and positive across four different estimations methods. Additionally, returns to education were also consistently positive and significant throughout the same period.

Overall, the results reported here are consistent with previous literature for Mexico and other countries supporting the notion of height as a measure of health capital that provides information on returns to health. A contribution from this analysis is the identification of differential returns by sex.

For example, in a recent estimation for individuals living in poverty in Mexico we identified returns to health in labor income that are also positive [10] Those estimates were produced using a different set of instrumental variables. The estimates reported here when instrumenting height are comparable in magnitude. Our estimations remain consistent in terms of both the sign and significance with previous findings for the ENSANUT 2006 dataset [9]. It is worth noting that the previous analysis included all adults 20 and older, whereas our current analysis focuses in individuals aged 25 to 45 [9]. The reported results align with positive returns to health documented in the ENSA 2000 dataset [30]. Similarly, analyses conducted on the Mexican population using different surveys yielded comparable estimates in terms of both sign and magnitude when employing the OLS specification. For instance, using data from 2002 & 2005, Vogl found increases ranging from1.4% to 2.3% in hourly earnings per 1 cm increase in height for males. In comparison, our estimates for 2000 and 2006 indicate increases of 2.8% and 2.6% for a 1% increase in height, roughly equivalent to a 1.7% and 1.6% increase per additional centimeter of height [31].

The returns identified in this analysis are consistent with those observed in other contexts. For instance, in Indonesia, an increase of 3.64% in income was estimated for each 1% increase in height. Similarly, in Pakistan, estimates suggest a roughly 1% increase in income associated with height, although this figure rises to 3% when height is instrumented. As in our case, the instrumental variable (IV) approach tends to yield larger returns. [5,32].

The increase in height between 2000 and 2018, as indicated, is small and is below expectation given the country's economic growth in the period [33]. Our analysis indicates a decrease in returns to health returns over that seems not related to changes in height as there are only small increases in height. A potential explanation may be related to the limited growth of the economy in general and of average earnings, as reflected in the almost null increase in the minimum wage in the period.

Compared to the OLS estimates, those generated using the Heckman correction show similar magnitudes, indicating that correcting for selection bias did not substantially alter the estimated returns. While the larger returns observed when instrumenting height may be attributed to limitations of the instrument, the primary conclusion from our analysis remains unchanged: there are significant and positive returns for health in Mexico. This conclusion holds true even in estimations without instruments, albeit with a lower magnitude.

Our analysis acknowledges the endogeneity of health and recognizes that this could affect the accuracy of the estimated returns to health. Provided that our instruments are adequate, the IV model is expected to yield more accurate estimates than the non-instrumented models. We explored alternative instruments with similar estimates to those reported. Specifically, children's height was utilized as an instrument for 2018, albeit with a smaller sample size and

exclusively for males due to data constraints. Although the estimates were slightly lower (5 compared to 8), a Wald test indicated no significant differences.

The unexpected negative returns for education among females in 2012 could be related to temporary dynamics in the labor market related to the non-skilled positions offered to women, but this is speculation on our part. It would be interesting to further explore this relationship in subsequent studies.

There are several limitations worthy noting. Firstly, the analyzed surveys were not designed to capture people's income accurately, and income is reported by a household informant and not from the person of interest directly, so our estimation could be biased. However, this limitation affects the four surveys equally, and there are no elements to assume that they are different over time in this potential bias, so the results between surveys would be maintained.

In addition, some differences in terms of population sampling were evident in the analysis. For the ENSANUT 2006, an oversampling of poor households seems to not be totally corrected (in the sense of assigning the real share of the population to those households) with the weighting, as there is a relatively lower socioeconomic status in average compared to the other surveys. In turn, for the ENSANUT 2018 anthropometric data were collected with a larger share of poor households than better off households, although this was partially corrected in the analysis by re-weighting observations to reproduce the survey distribution of socioeconomic status.

## Conclusion

The positive health returns identified are consistent with the notion that an effective mechanism to promote social mobility is to invest in health and nutrition during childhood and adolescence, that is, during periods of growth.

Mexico is a country with high social inequality, which represents a central challenge for the country's development. In this sense, identifying mechanisms that favor upward social mobility is imperative to close the gaps in the population. Focusing on the health of infants and adolescents is not only valuable for the intrinsic value of health in general and the future generations, but also as a mechanism to reduce inequality, that is, investments in health, in the phases of greater human capital formation are an essential catalyst for social mobility.

## Author Contributions

**Conceptualization:** Juan Pablo Gutiérrez, Stefano M. Bertozzi.

**Data curation:** Juan Pablo Gutiérrez.

**Formal analysis:** Juan Pablo Gutiérrez.

**Methodology:** Juan Pablo Gutiérrez.

**Writing – original draft:** Juan Pablo Gutiérrez, Stefano M. Bertozzi.

**Writing – review & editing:** Stefano M. Bertozzi.

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
