## [Decision Letter · Decision Letter 0]

6 Feb 2024

PONE-D-23-39002Height & income: labor returns of health in Mexico from 2000 to 2018PLOS ONE

Dear Dr. Gutierrez,

Thank you for submitting your manuscript to PLOS ONE. After careful consideration, we feel that it has merit but does not fully meet PLOS ONE’s publication criteria as it currently stands. Therefore, we invite you to submit a revised version of the manuscript that addresses the points raised during the review process.

**The document discusses an in interesting topic, but the referees have major comments that need to be address; in my opinion, most of them are clarifications which will greatly improve the quality of the document.**

We look forward to receiving your revised manuscript.

Kind regards,

Paul Rodríguez-Lesmes, Ph.D.

Academic Editor

PLOS ONE

Journal Requirements:

Additional Editor Comments:

Please include a letter with your response to each of the points presented by the reviewers below.

Reviewers' comments:

Reviewer's Responses to Questions

**Comments to the Author**

1. Is the manuscript technically sound, and do the data support the conclusions?

Reviewer #1: Yes

Reviewer #2: Partly

2. Has the statistical analysis been performed appropriately and rigorously? 

Reviewer #1: Yes

Reviewer #2: No

3. Have the authors made all data underlying the findings in their manuscript fully available?

Reviewer #1: Yes

Reviewer #2: Yes

4. Is the manuscript presented in an intelligible fashion and written in standard English?

Reviewer #1: Yes

Reviewer #2: Yes

5. Review Comments to the Author

Reviewer #1: I find the article interesting. While the topic has been already studied for Mexico and other countries, the contribution of the authors is still important for developing economies. I have some minors comments that might improve the document:

1. Are the 4 surveys comparable? I think that the questionnaire of the 2000 survey is different, and I am not sure if it is comparable to the others.

2. Are the anthropometric measures taken by qualified personnel or self-reported by respondents. Please clarify.

3. It seems that it is a coauthored paper, but in some sentences, it is referred as, for example: “In addition, I generated a socioeconomic status index of the household…”

4. The authors use individuals between 25-45 y/o to focus on those whose height is the maximum possible and has not yet beginning to decline. I think this selection needs a reference. Please add a reference.

5. Do you individuals with income equal to zero in your sample? If so, how do you treat this cases when applying the natural logarithm?

6. Analysis section: “We estimated experience as age minus the schooling years and minus 6 years, as suggested in previous studies”, please specify which studies. Add the references.

7. The results for the education coefficient are somehow surprising (sometimes negative). The authors only describe the results but do not provide a possible explanation or discussion. I think a discussion on this is needed.

Reviewer #2: The article presents an analysis of the potential association between height and income over time in Mexico. However, although the results are interesting, there are several things that need to be considered before publication.

Abstract:

This sentence needs to be revised using adults' height as an indicator of the individual's health capital accumulation. Previous analyses have reported inconsistent estimates of the health returns in Mexico, although they use different approaches. Which approaches? And compared to what?

In addition, the purpose of the paper in the abstract is not clear.

Introduction:

The introduction should be written in the third person.

Methodology:

It should clearly state the data sources, the sampling method of each data source, and the sample size. Also, it is important to clearly present the methods of generating variables such as income, it is not clear if it was an imputation or if the authors created a wealth index. This information should be provided in the text; if necessary, additional information should be included as supplementary material.

Please use subheadings to present the data, variables and methods.

It Will be ideal that the authors present a more detailed description of the different methods, and how they harmonized the data to make all surveys comparable across years.

Results

It is necessary that authros explain better why the percentage of men across time has reduced. Most surveys collect information from all household members, not only from the person who is at the house. Therefore, depending on the data collection process, the survey Will or may not be representative of individuals. A SEntence such as "is common to have a larger percentage of women as they are more likely to be at their dwellings, a fact that is confirmed by the differential in labor participation" only applys if the survey only collects information from one main respondent and it does not collect information of other household members. Please explain this better, because for example ENSANUT is representative of members of private households, therefore men and women (regardless of whether they are at home when the survey was collected).

If the sample between 2000 and 2006 was not comparable, why did you only see this on the education indicators? It is necessary to explain this better and include references supporting the argument that the oversampling of poor areas did not correct sampling weights.

Table 1 needs to be presented better; in its current form, it is confusing and not easy to read. Also percentages should be presented as percentages not as a fractions.

Can you please include CI in figure 1. Also it will be better if you include in the X axis the years, and each of the lines represents a decile, in that way it is possible to see the gap better.

Also, please consider that line graphs are not used to present decides. In that case use bar grapsh

Table 2 is confusing and needs to be presented better,

Authors need to explain better why the education variable behaves so differently between models and what are the implications of this, given that education is a stronger explanatory variable, this should be explained in detail and a more comprehensive analysis should be conducted.

One main problem that I find in the article is that the authors give the sense that an increase in height increases income, but there is not a significant increase in height over time on average, and income does not increase either. In addition, the article does not provide enough details to understand if the increase in height for people in the highest decile was larger than in the lowest decile, or at least how the data is presented it is not possible to understand if the differences are significant and increase over time. Most importantly, the data is not comparable over time, or at least this is not clear in the methods section (does 2000 data uses the same sampling frame as 2018?) therefore, it is not clear if the analysis can be conducted with this data, and if the results are showing real differences over time, or just differences in the sampling frame. Finally, the results of the four estimation models are not consistent; this is not properly explained in the article.

The discussion does not present limitations and the discussion needs to be improved, including potential explanations for the results.

The changes in the education results for men and women are difficult to explain.

The article needs to improve the way tables and graphs are presented.

Finally the article should be copy-edited.

6. PLOS authors have the option to publish the peer review history of their article (what does this mean?). If published, this will include your full peer review and any attached files.

Reviewer #1: No

Reviewer #2: No

---

## [Decision Letter · Decision Letter 1]

5 Apr 2024

PONE-D-23-39002R1Height & Income: Analyzing the Labor Income Returns to Health in Mexico from 2000 to 2018PLOS ONE

Dear Dr. Gutierrez,

Thank you for submitting your manuscript to PLOS ONE. After careful consideration, we feel that it has merit but does not fully meet PLOS ONE’s publication criteria as it currently stands. Therefore, we invite you to submit a revised version of the manuscript that addresses the points raised during the review process.

**Both reviewers are ok with your revisions, but both consider that you perform "a final revision of the language and the Journal guidelines about tables and figures". Please follow their suggestion in order to proceed with the publication of the paper. Great job!**

We look forward to receiving your revised manuscript.

Kind regards,

Paul Rodríguez-Lesmes, Ph.D.

Academic Editor

PLOS ONE

Journal Requirements:

Additional Editor Comments:

Please follow Reviewer 2 suggestions

Reviewers' comments:

Reviewer's Responses to Questions

**Comments to the Author**

1. If the authors have adequately addressed your comments raised in a previous round of review and you feel that this manuscript is now acceptable for publication, you may indicate that here to bypass the “Comments to the Author” section, enter your conflict of interest statement in the “Confidential to Editor” section, and submit your "Accept" recommendation.

Reviewer #1: All comments have been addressed

Reviewer #2: All comments have been addressed

2. Is the manuscript technically sound, and do the data support the conclusions?

Reviewer #1: Yes

Reviewer #2: Yes

3. Has the statistical analysis been performed appropriately and rigorously? 

Reviewer #1: Yes

Reviewer #2: Yes

4. Have the authors made all data underlying the findings in their manuscript fully available?

Reviewer #1: Yes

Reviewer #2: No

5. Is the manuscript presented in an intelligible fashion and written in standard English?

Reviewer #1: Yes

Reviewer #2: Yes

6. Review Comments to the Author

**Reviewer #1: **I find the article almost ready for publication.

I recommend a final revision of the language and the Journal guidelines about tables and figures.

The authors successfully addressed all my concerns.

**Reviewer #2: **Please edit and present bettert the tables, in the current state is really difficult to follow and understand. Also, please check that you the evidence you are presenting does not support the ideat that "are there is a robust positive

relationship between height and socioeconomic decile.", only in extreme cases there are significant differences between hight. Also explian and present the robustness analysiss. Finally, you are presenting three models, each reach to a similar conclusion, but still in some models the coefficient of years of schooling is not expected (e.g. 2012) and the difference between the models vs the IV models is large, please explain why and which model is the best. Did you conduct any comparison between the thre models

7. PLOS authors have the option to publish the peer review history of their article (what does this mean?). If published, this will include your full peer review and any attached files.

Reviewer #1: No

Reviewer #2: No

---

## [Author Response · Author response to Decision Letter 1]

16 Apr 2024

We addressed the comments in the manuscript and attached a file with the answers to each comments.

---

## [Editor Report · Decision Letter 2]

19 Apr 2024

Height & income: labor returns of health in Mexico from 2000 to 2018

PONE-D-23-39002R2

Dear Dr. Gutierrez,

We’re pleased to inform you that your manuscript has been judged scientifically suitable for publication and will be formally accepted for publication once it meets all outstanding technical requirements.

Kind regards,

Paul Rodríguez-Lesmes, Ph.D.

Academic Editor

PLOS ONE

---

## [Editor Report · Acceptance letter]

29 Apr 2024

PONE-D-23-39002R2 

PLOS ONE

Dear Dr. Gutierrez, 

I'm pleased to inform you that your manuscript has been deemed suitable for publication in PLOS ONE. Congratulations! Your manuscript is now being handed over to our production team.

Kind regards, 

on behalf of

Mr Paul Rodríguez-Lesmes 

Academic Editor

PLOS ONE